# Deep Reinforcement Learning-Based Scheduler on Parallel Dedicated Machine Scheduling Problem towards Minimizing Total Tardiness

Donghun Lee, Hyeongwon Kang, Dongjin Lee, Jeonwoo Lee and Kwanho Kim *

Department of Industrial and Management Engineering, Incheon National University, Incheon 22012, Republic of Korea
* Correspondence: khokim@inu.ac.kr

**Abstract:** This study considers a parallel dedicated machine scheduling problem towards minimizing the total tardiness of allocated jobs on machines. In addition, this problem comes under the category of NP-hard. Unlike classical parallel machine scheduling, a job is processed by only one of the dedicated machines according to its job type defined in advance, and a machine is able to process at most one job at a time. To obtain a high-quality schedule in terms of total tardiness for the considered scheduling problem, we suggest a machine scheduler based on double deep Q-learning. In the training phase, the considered scheduling problem is redesigned to fit into the reinforcement learning framework and suggest the concepts of state, action, and reward to understand the occurrences of setup, tardiness, and the statuses of allocated job types. The proposed scheduler, repeatedly finds better Q-values towards minimizing tardiness of allocated jobs by updating the weights in a neural network. Then, the scheduling performances of the proposed scheduler are evaluated by comparing it with the conventional ones. The results show that the proposed scheduler outperforms the conventional ones. In particular, for two datasets presenting extra-large scheduling problems, our model performs better compared to existing genetic algorithm by 12.32% and 29.69%.

**Keywords:** machine scheduling; deep reinforcement learning; parallel dedicated machines; sustainable manufacturing; total tardiness objective

## 1. Introduction

The parallel dedicated machine scheduling (PDMS) problem is of paramount importance both for academic researchers and practitioners, as scheduling is getting important, especially in large-scale industrial manufacturing, such as shipbuilding and aircraft [1,2]. In particular, since this scheduling problem is associated with the photolithography stage, which is a key point for enhancing productivity in semiconductor manufacturing, an effective scheduling method has been recently required [3]. Unlike a parallel machine scheduling problem, the jobs processed by a machine are limited according to job types in a PDMS problem [4]. A setup task is necessary on a machine before it processes a job with a different job type compared to that of a job processed just before on the machine.

For the PDMS problem considered in this study, each job is associated with a job type, processing time, and due date, and it can be processed by only one of the dedicated machines to its job types defined in advance. A dedicated machine is able to process one job at a time at most.

It is a challenging problem to develop a scheduling method to minimize the tardiness and the number of setups of jobs at the same time. The minimization of the total tardiness objective is a well-known NP-hard scheduling problem [5] and is often considered the most important performance measure in the manufacturing environment since the products should be released before their due dates, which are set by customers [6]. Allocations of jobs on machines that only consider minimizing tardiness may utilize whole machines and tend to process jobs with various types. In that case, the available capacity loss of machines

increases from 20 percent to 50 percent, which caused by unnecessary setups [7]. On the contrary, as available machines in terms of setup state are not enough at a particular time, jobs with identical job types might be allocated on a limited number of machines, which results in the tardiness of these allocated jobs and is likely to be increased eventually.

Several priority rule-based algorithms have been studied to minimize total tardiness for parallel machine scheduling problems [4,8–10]. Recently, a variable neighborhood search algorithm was applied to search for a schedule to minimize total weighted tardiness in a parallel machine scheduling problem with eligibility constraints, and it is better than a mixed integer linear programming and existing models [11]. However, the scheduling performances of these algorithms are not guaranteed in a real manufacturing environment since they simply consider the preference of jobs and machines in a myopic manner. In the meantime, meta-heuristics, such as the genetic algorithm, tabu search, ant colony, and simulated annealing, were proposed [12–15]. In addition, a hybrid evolutionary algorithm has been recently proposed to resolve a parallel machine scheduling problem with resources constrained with setup times [16]. Although such methods successfully search for a near-optimal schedule, they may also not be practical due to the exhaustive computation time for yielding schedules.

Some supervised learning-based schedulers using a neural network (NN) were suggested to yield a superior schedule by exploring wider solutions in the training phase [17]. For the parallel machine scheduling problem with sequence-dependent setups that have different setup times depending on the pair of a job type and a machine [18], an NN-based scheduler was proposed, and its scheduling performances outperform the heuristic schedulers by 6%. Although NN-based schedulers are much more effective compared to heuristic schedulers in a fast manner, the obtained schedule may not be satisfactory since a neural-network-based scheduler only uses the training instances acquired from a simple heuristic scheduler.

In recent years, Q-learning techniques have been adopted to obtain high-quality schedules in terms of minimizing total tardiness for parallel scheduling problems [19,20]. Q-learning-based schedulers are designed to choose one of the priority rules as an action that determines the allocation of a job on a machine. They exhibit better performances compared to the several heuristic schedulers. More recently, deep Q-network-based schedulers were suggested for semiconductor manufacturing applications [21,22]. A deep deterministic policy gradient (DDPG)-based scheduler was also proposed to minimize weighted tardiness in the stochastic parallel machine scheduling problem [23]. Unlike [19,20], these studies developed multi-agent approaches where each agent considers the allocation of a job on a machine, and they successfully improved performances by reducing the learning complexity.

However, such methods still have some limitations when solving the scheduling problem considered in this paper. The well-known overestimation problem of deep Q-network causes allocation uncertainty on one of the dedicated machines or increasing tardiness of allocated jobs [24]. Designing actions based on widely known priority rules is also difficult considering the allocation of the jobs on one of the dedicated machines due to allocation uncertainty problem. The allocation uncertainty indicates that a machine is not able to accommodate job types that are not being pre-designed in advance.

Therefore, to overcome the limitations mentioned above, we newly propose a scheduler, called the DDQN scheduler, based on double deep Q-learning to address the PDMS problem with a sequence-independent setup time to minimize the total tardiness of allocated jobs. The proposed scheduler searches for the allocation patterns between jobs and machines to minimize the tardiness of a job for the PDMS problem during the training.

Specifically, a state is designed to represent the relationships between the possible allocations of candidate jobs and machines in terms of occurrences of setup and tardiness. By using this state design, the proposed model is capable of determining allocation based on how much this allocation possibly contributes to the tardiness caused by candidates in the future. The reward is set to be a negative value based on tardiness and the allocation possibility of jobs on machines to both minimize tardiness and ensure the allocation of

jobs to machines. The training algorithm is also designed to stably learn by overcoming an infeasible action problem [25].

Some novelties of our study and the research gap are the following:

(1) A new state and reward are designed to capture the allocation impact for the candidates in the future and ensure the allocation of jobs on machines in the PDMS problem;

(2) Unlike existing deep Q-learning methods, the proposed scheduler is able to successfully train by overcoming an infeasible action selection problem causing an overfitting problem by the suggested training algorithm;

(3) The numerical experiments are conducted by using eight datasets consisting of small, medium, large, and extra-large scheduling problems demonstrate that the proposed scheduler outperforms the previous ones in terms of total tardiness. In particular, for the dataset including extra-large scheduling problems, our scheduler obtains quite effective performances compared to existing scheduling methods.

The rest of this article is organized as follows. In Section 2, the considered PDMS problem is defined. Section 3 introduces the proposed framework, the training, and the scheduling phases. Next, numerical experiments are conducted to compare the scheduling performances between the proposed scheduler and the conventional schedulers. Finally, in Section 5, the conclusion and some future directions are discussed.

## 2. Problem Definition

A PDMS problem composed of $N_J$ jobs, where the $j^{th}$ job is denoted as $J_j$. $J_j$ consists of job type $e_j$, processing time $p_j$, and due date $d_j$. $U_j$ is a set of dedicated machines that can allocate job $J_j$. Each job is processed by one of $N_M$ machines where the $i^{th}$ machine is denoted as $M_i$.

When the processing of job $J_j$ is completed on machine $M_i$, its completion time is denoted as $c_j$. Furthermore, when the processing of a job on a machine is completed after its due date, the tardiness of the job occurs. Specifically, the tardiness of job $J_j$ on machine $M_i$, $\tau_j$ is defined as Equation (1). Finally, for a PDMS problem, the total tardiness of a schedule is calculated by summing each tardiness of allocated jobs obtained by using Equation (1).

$$\tau_j = max(c_j - d_j, 0) \tag{1}$$

The mathematical formulation is provided to explain the PDMS considered more clear based on the previous research [26]. Two decision variables, such as $x_{i,l,j}$ and $x_{i,0,j}$, are required. If job $J_j$ is processed after job $J_l$ on machine $M_i$, $x_{i,l,j}$ is one, otherwise, it is zero. If job $J_j$ is the first or last job to be processed on machine $M_i$, $x_{i,0,j}$ is one, otherwise is zero.

Minimize $\sum_{j=1}^{N_J} \tau_j$

Subject to

$$\sum_{\substack{l=0 \\ l \neq j}}^{n} \sum_{i \notin U_j}^{m} x_{i,l,j} = 1, \ \forall j = 1, 2, \ldots, N_J \tag{2}$$

$$\sum_{\substack{l=0 \\ l \neq j}}^{n} \sum_{i \notin U_j}^{m} x_{i,l,j} = 0, \ \forall j = 1, 2, \ldots, N_J \tag{3}$$

$$\sum_{\substack{l=0 \\ l \neq b}}^{n} x_{i,l,b} - \sum_{\substack{j=0 \\ j \neq b}}^{n} x_{i,b,j} = 0, \ \forall b = 1, 2, \ldots, N_J \ and \ i = 1, 2, \ldots, N_M \tag{4}$$

$$c_j \geq c_l + \sum_{i=1}^{m} x_{i,l,j} \cdot \left( s_{l,j} + p_j \right) + T \cdot \left( \sum_{i=1}^{m} x_{i,l,j} - 1 \right), \forall l = 0, 1, \ldots, N_J \ and \ j = 1, 2, \ldots, N_J \tag{5}$$

$$\sum_{j=0}^{n} x_{i,0,j} = 1, \ \forall i = 1, 2, \ldots, N_M \tag{6}$$

$$x_{i,l,j} \in \{0, 1\}, \ \forall l = 0, 1, \ldots, N_J, \ j = 0, 1, \ldots, N_J, \ and \ i = 1, 2, \ldots, N_M \tag{7}$$

$$c_0 = 0 \tag{8}$$

$$c_j \geq 0, \ \forall j = 1, 2, \ldots, N_J \tag{9}$$

The objective is to minimize the total tardiness of allocated jobs on machines. Constraints (2) and (3) restrict that each job should be processed on a dedicated machine. Each job must neither be preceded by more than one job according to Constraint (4). From Constraint (5), the completion time of a job on each machine must be at least larger than or equal to the sum of completion time of the previous job, setup time between two jobs, and processing time of the job. *T* is very large value. Constraint (6) restricts the number of jobs processed, only one job should be processed first at each machine. Constraints (7–9) indicate the conditions of decision variables. Particularly, Constraint (9) initializes the completion time of a dummy job to be zero.

## 3. Proposed Machine Scheduler

### 3.1. Research Framework

The differences between deep learning and reinforcement models are the following points. First, the deep learning model requires many training instances, but the reinforcement learning model is able to train without them. Second, the deep learning model utilizes a supervised learning mechanism to find hidden patterns between input and output values, but reinforcement learning trains through trial and error by searching for a policy able to obtain maximized rewards. Finally, while the design of input and output values is required for the training of a deep learning model, state, action, and reward are needed for the training of a reinforcement learning model.

The overall framework of the proposed scheduler, which is based on double deep Q-learning, is presented in Figure 1. It consists of two phases, the training and scheduling phases. In the training phase, many PDMS problems each of which contains $N_J$ jobs with job types, processing times, and due dates are prepared. The initial setup status of each machine is set randomly.

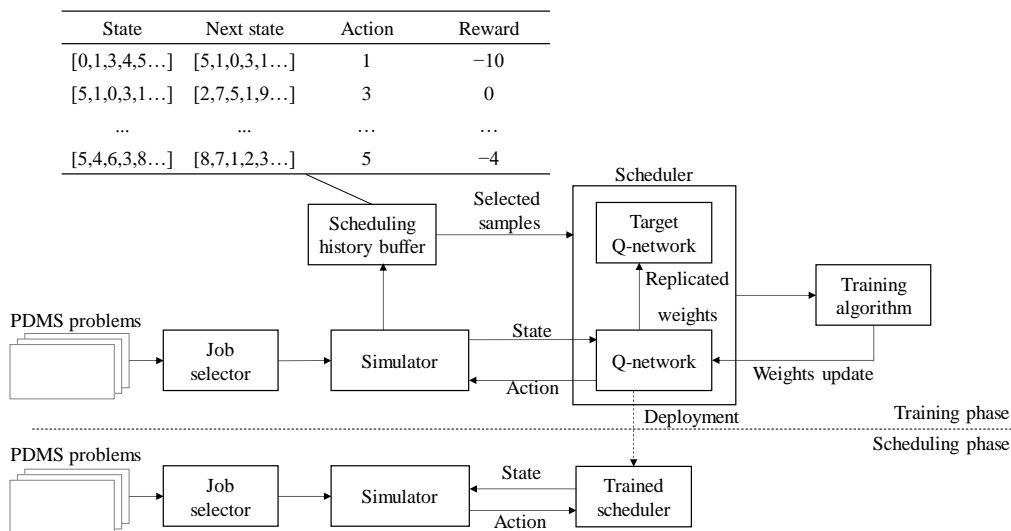

**Figure 1.** Overall framework of the proposed scheduler.

In the training phase, for a given PDMS problem, the job selector chooses a job by using the heuristic rules combined with the least slack time (Slack) and the earliest due date (EDD) is applied to simply determine the selected job for each time [27]. A simulator plays an environment role in the proposed framework. Whenever a job is selected by the job selector, the simulator provides a state to the Q-network of the proposed scheduler.

Then, the simulator receives an action determined by the Q-network and allocates the job on a machine based on the action.

After allocation, the next state and a reward are observed. The reward is designed to consider both the allocation preference of a job on a machine and its possible tardiness. If the allocation of a job on a machine is infeasible, the reward is calculated based on both this status and the tardiness of the allocated job; otherwise, it is obtained based on only tardiness. The set of allocation transitions, which consist of state, next state, action, and reward, are stored in a scheduling history buffer and utilized to alleviate the correlations for training purposes [28].

Several sets of allocation transitions are randomly selected. By using the selected allocation transitions, the Q-network of the proposed scheduler attempts to search for the optimized weights in terms of the rewards by gradually updating and cyclically replicating the target Q-network of the proposed scheduler as suggested in [29].

In the scheduling phase, the trained Q-network of the proposed scheduler is only utilized without the target Q-network. Given a PDMS problem, this scheduler acts to allocate each selected job by the job selector on one of the machines. The schedule is obtained after finalizing the allocations of all the jobs considered, and the performance of each schedule is then evaluated in terms of total tardiness.

### 3.2. State, Action, and Reward

A state consists of the details of the current snapshot regarding to candidate jobs and machines, as shown in Table 1. Here, the state is designed to capture the potential impact caused by all the possible pairs of $k$ candidate jobs and $N_M$ machines. This state is beneficial to accurate observation and the action of maximizing reward since it contains all the combinations of pairs between $k$ candidate jobs and $N_M$ machines in terms of setup, allocation possibility, and expected tardiness.

**Table 1.** Features of a state.

| Notation | Descriptions | Dimension |
|---|---|---|
| $f_{k,N_M}$ | Occurrences of setups between $k$ candidate jobs for $N_M$ machines | $\mathbb{R}^{k \times N_M}$ |
| $v_{k,N_M}$ | Statuses of allocations between $k$ candidate jobs and $N_M$ machines | $\mathbb{R}^{k \times N_M}$ |

Specifically, the first feature in Table 1 represents whether a setup occurs or not for each pair of a job and a machine comprising $k$ candidate jobs and $N_M$ machines as $f_{k,N_M}$. If a setup is conducted, its value is one, otherwise, it is zero. This feature enables the proposed scheduler to understand a possible setup expected in the next step when a particular allocation is determined. Next, the allocation availabilities for each pair of $k$ candidate jobs and $N_M$ machines are considered and denoted as $v_{k,N_M}$. Based on this feature, the proposed scheduler captures possible allocations of $k$ candidate jobs and $N_M$ machines. Finally, the expected tardiness for each pair of $k$ candidate jobs and $N_M$ machines is utilized and denoted as $u_{k,N_M}$. The feature helps the proposed scheduler understand possible occurrences of tardiness when a particular allocation is conducted. The dimension of each feature is $\mathbb{R}^{k \times N_M}$.

For job $J_j$, its state is denoted as state $s_j$, the possible action set for the state is defined as $A(s_j)$, where $a_i \in A(s_j)$ means that machine $M_i$ is selected for the job. After the allocation is conducted, next state $s_{j+1}$ for a job considered in the next step and reward $r_j$ are updated. Finally, $s_j$, $s_{j+1}$, $a_i$, and $r_j$ are stored as a set of allocation transitions in the scheduling history buffer.

Reward $r_j$ is designed to consider allocation possibility and minimize the tardiness of a job, defined as follows:

$$r_j = \begin{cases} -\tau_j - T, & J_j \notin U_j \\ -\tau_j, & otherwise \end{cases} \tag{10}$$

The total reward is defined by the summing up of the rewards of all the jobs considered, as follows:

$$R = -\left(\sum_{j=1}^{N_J} \sum_{i=1}^{N_M} \tau_j + T\right) \tag{11}$$

Note that maximizing $R$ is equal to minimizing the rewards of $N_J$ jobs on $N_M$ machines [28].

### 3.3. Archetecture of the Proposed Scheduler

A deep neural network consisting of an input, five hidden, and an output layers in a Q-network is applied, as shown in Figure 2. The input values of the proposed Q-network are a state as presented in Table 1. The output value is the predicted Q-value for an allocation of a job on a machine. The proposed scheduler is able to produce a schedule regardless of any number of jobs and processes since it is independently related to the job and process.

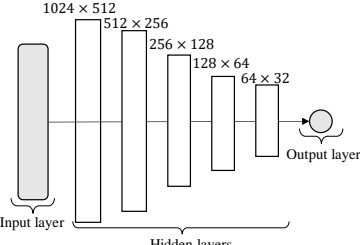

**Figure 2.** Q-network structure of the proposed scheduler.

Although a Q-network of the DDQN method is similar to the DQN method, the DDQN method is of more benefit than the DQN method when a model trains since it is successful to prevent the overestimation problem for the action values [30]. Therefore, the proposed scheduler is able to stably train for the considered scheduling problem through the training algorithm of the DDQN method.

In detail, the rectifier linear unit is applied in each hidden layer as an activation function to overcome the gradient vanishing and radiant exploding problems [31]. Linear units are used in the output layer of both the proposed Q-network and target Q-network since the output value is approximating a negative value for each action. RMSProp is adopted as an optimizer to adjust the weights of the proposed Q-network during the training [32].

In the reinforcement learning mechanism, the training performances may change according to hyper-parameter settings such as the number of neurons, learning rate, and scheduling history buffer size. Accordingly, the random search is implemented to find optimal hyper-parameters that yield the best performances [33]. The details for hyper-parameter settings are introduced in Section 4.

### 3.4. Training and Scheduling Phase

The overall training process of the proposed scheduler is conducted by using Algorithm 1. Through Algorithm 1, the proposed scheduler is successful in training by overcoming the infeasible action problem. We prepared a Q-network with random weights $w$, target Q-network with weights $\hat{w}$, and scheduling history buffer $B$ with the size of $g$. Each scheduling problem is considered an episode, and the algorithm is repeatedly conducted until $\sigma$ episodes.

---

**Algorithm 1** Training procedure of the proposed scheduler

---

Input: PDMS problem
Output: Q-network
1: Initialization: Q-network with random weight $w$, target Q-network with weight $\hat{w} = w$, the number of episodes $\sigma$, and scheduling history buffer $B$ with size $g$
2: for each $o$ in $\sigma$ do
3:      execute job selector
4:      for each $j$ in $N_J$ do
5:           calculate $s_j$
6:           get $a_i$ and machine $i$ based on Algorithm 2
7:           if $a_i$ is infeasible then
8:                get $r_j$
9:                $s_{j+1} = s_j$
10:              store set $(s_j, a_i, r_j, s_{j+1})$ in $B$
11:              select sets randomly $(s_n, a_n, r_n, s_{n+1}) \in B$
12:              calculate $q_n = Q(s_n, a_n; w)$
13:              calculate $y_n = r_n + \gamma \underset{a}{\text{argmax}}\ \hat{Q}\left(s_{n+1}, \underset{a}{\text{argmax}} Q(s_{n+1}, a; w); \hat{w}\right)$
14:              calculate loss L from Equation (12)
15:              update weight $w$ by RMSProp and L
16:              $j \leftarrow j + 1$
17:         Else
18:              get $r_j$ and $s_{j+1}$
19:              store set $(s_j, a_i, r_j, s_{j+1})$ in $B$
20:              select sets randomly $(s_n, a_n, r_n, s_{n+1}) \in B$
21:              calculate $q_n = Q(s_n, a_n; w)$
22:              calculate $y_n = r_n + \gamma \underset{a}{\text{argmax}}\ \hat{Q}\left(s_{n+1}, \underset{a}{\text{argmax}} Q(s_{n+1}, a; w); \hat{w}\right)$
23:              calculate loss L from Equation (12)
24:              update weight $w$ by RMSProp and L
25:     end for
26:     Replicate $\hat{w}$ from $w$
27: end for
28: return Q-network

---

Given a PDMS problem, the proposed job selector chooses a job by considering the priorities of jobs (line 3). Then, the allocation task is conducted based on state $s_j$ by using Algorithm 2 that determines an action with the $\varepsilon$-greedy strategy, which is broadly applied in deep Q-learning [22,34]. This $\varepsilon$-greedy strategy is able to ensure an active distribution of the adequate exploration during the training of the proposed scheduler [35]. In Algorithm 2, if random value $z$ is lower than epsilon value $\varepsilon$, action $a_i$ is randomly selected; otherwise, action $a_i$ is determined based on the maximum Q-value for action set $A(s_j)$ (lines 3–6).

---

**Algorithm 2** Allocation decision with $\varepsilon$-greedy strategy

---

Input: State $s_j$
Output: Action $a_i$ and machine $M_i$
1: Sample a random value $z \in [0, 1]$
2: If $z < \varepsilon$ then
3:      Select action $a_i$ randomly from $A(s_j)$
4:      $\varepsilon = \varepsilon * \theta$
5: Else
6:      $a_i = \left(\underset{a \in A(s_j)}{\text{argmax}} Q(s_j, a; w)\right)$
7: Return Action $a_i$ and machine $M_i$

---

Based on the current allocation status calculated from action $a_i$, it is checked whether this action is feasible or not. If action $a_i$ for job $J_j$ is infeasible, the proposed Q-network and target Q-network train until they find the weights able to select the feasible action for job $J_j$ while considering the tardiness (lines 7–16). Otherwise, they train by the typical training algorithm of the DDQN method (lines 18–24). In lines 10 and 19, the size of sets is checked on whether it is $g$ or not. If it equals size $g$, a new transition set replaces the oldest transition set. To update weights $w$, $n$ transition sets are randomly selected, where $n$ is the number of transition sets, and loss value $L$ is calculated using Q-value $q_n$ and target value $y_n$ (lines 11–15 and lines 20–24). Here, $\gamma$ is defined as a discount factor and used to consider the uncertainty of future allocation [36]. In detail, mean squared error is used for loss value $L$, which is calculated by using Equation (12).

$$L = \frac{1}{n} \sum_{n=1}^{n} (q_n - y_n)^2 \tag{12}$$

Based on loss value $L$, the RMSProp optimizer adjusts weight $w$ for $n$ transition sets. Whenever an episode is completed, weights $\hat{w}$ of the target Q-network are updated by replicating weights $w$ of the Q-network for ensuring stable training [29]. The training phase is completed after processing $\sigma$ episodes, and the trained scheduler that exhibits the best scheduling performances in terms of total tardiness is then used in the scheduling phase.

In the scheduling phase, based on the trained scheduler, only lines 3–6 in Algorithm 1 are utilized to produce a schedule for a PDMS problem. The allocation of a job for a machine is decided by the action, which is determined based on the maximum Q-value without random action selection in Algorithm 2.

## 4. Experiments

### 4.1. Experiment Settings

Eight datasets were prepared varying the number of jobs, the range of processing times, and the range of due dates, as shown in Table 2. The processing times and due dates for the jobs are determined by the uniform distribution. The problem size of each dataset was set to be small, medium, large, and extra-large according to the number of jobs. The number of job types and machines was 13 and 18, respectively, for the all datasets. Setup time was 2 regardless of a pair of a job type and a machine. Experiments were conducted on a Ryzen 3900X-3.8-GHz (AMD, Santa Clara, CA, USA) PC with 32-GB memory (manufacturer, location) and GPU-2080 (NVIDIA, Santa Clara, CA, USA).

**Table 2.** Datasets used in this study.

| Datasets | Number of Jobs | Distribution of Processing Times | Distribution of Due Dates |
|---|---|---|---|
| 1 | 30 | (2,6) | (2,8) |
| 2 | 50 | (2,6) | (2,12) |
| 3 | 70 | (2,6) | (2,16) |
| 4 | 100 | (2,6) | (2,25) |
| 5 | 30 | (4,10) | (4,14) |
| 6 | 50 | (4,10) | (4,25) |
| 7 | 70 | (4,10) | (4,39) |
| 8 | 100 | (4,10) | (4,50) |

To examine the effectiveness of the DDQN scheduler, conventional rule-based schedulers, such as Slack-EDD, apparent tardiness cost with setup (ATCS), and cost over time (COVERT) were implemented for reasons of comparison. These rules show good scheduling performances to minimize total tardiness [37,38]. Additionally, deep Q-learning- and meta-heuristic-based schedulers, named TPDQN [21] and GAS [39], were implemented to compare the scheduling performances. The state and reward of TPDQN were designed to be equal to those of the proposed scheduler, but the hyper-parameters and training

algorithm were the same as it in [21]. For the development of GAS, the hyper-parameters settings, i.e., the number of generations, the ratios of mutation and crossover, were set to be identical to the original version. Although a schedule in the real-world environment is typically generated on an hourly basis, GAS was set to terminate within one hour [40].

The scheduling performances between schedulers were relatively compared by using relative percentage deviation (*RPD*) defined as Equation (13), where $OBJ_{best}$ and $OBJ$ are the best schedule for a scheduling problem in terms of total tardiness and the total tardiness obtained by the considered scheduler, respectively [41]. The results 0 and 100 were the best and the worst, respectively.

$$RPD = \frac{OBJ - OBJ_{best}}{OBJ_{best}} \times 100 \tag{13}$$

*4.2. Hyper-Parameter Settings*

To determine the best hyper-parameters, a random search was conducted according to the number of hidden layers and neurons, learning late, and scheduling buffer size *B* by referring to [28]. Other hyper-parameters included in the number of episodes $\sigma$, optimizer, loss function, decay factor $\theta$, discount factor $\gamma$, and epsilon value $\varepsilon$ were identical to the set by 2000, RMSProp, mean squared error, 0.999, 0.99, and 1.0, respectively, through repeat experiment trials.

Figure 3a-for the structures of a network (500, 200) and (1024, 256, 32), total tardiness changes sensitively until 100 episodes, but it becomes similar after 1000 episodes. The total tardiness of others continuously decreases.

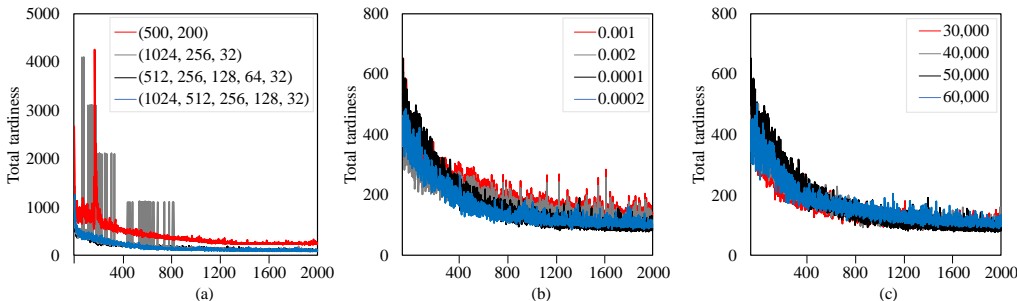

**Figure 3.** Changes of total tardiness according to hyper-parameters settings: (**a**) Number of hidden layers and neurons; (**b**) Learning late; (**c**) Scheduling buffer size.

Figure 3b,c indicates that total tardiness tends to gradually decrease regardless of parameter settings. When the learning rates and scheduling buffer sizes are high, the training performances are much better. The performance difference between learning rates 0.0001 and 0.0002 and between buffer sizes 50,000 and 60,000 are not significant, respectively. Yet, when learning late 0.0002 and buffer size 50,000 are applied, the best training performance is observed. Therefore, based on the results, we determined the structure of a network, learning rate, and scheduling buffer size by (1024, 512, 256, 128, 32), 0.0002, and 50,000, respectively.

*4.3. Experiment Results*

The scheduling performances of the schedulers considered in this study were compared in terms of total tardiness for each dataset. Figure 4 shows *RPD* achieved by all the schedulers for each dataset, and Table 3 summarizes the average *RPD* of each model. The proposed scheduler provides better scheduling performances than the other schedulers for most datasets. It means that our model is successful in providing schedules able to minimize total tardiness regardless of the number of jobs and process. In particular, this scheduler outperforms the others as the problem size becomes bigger, as shown in Figure 4 and Table 3. Among the baselines, GAS shows outstanding scheduling performances and

gives better scheduling performances compared to the proposed scheduler for Datasets 1 and 6. The scheduling performances of ATCS are slightly better than COVERT.

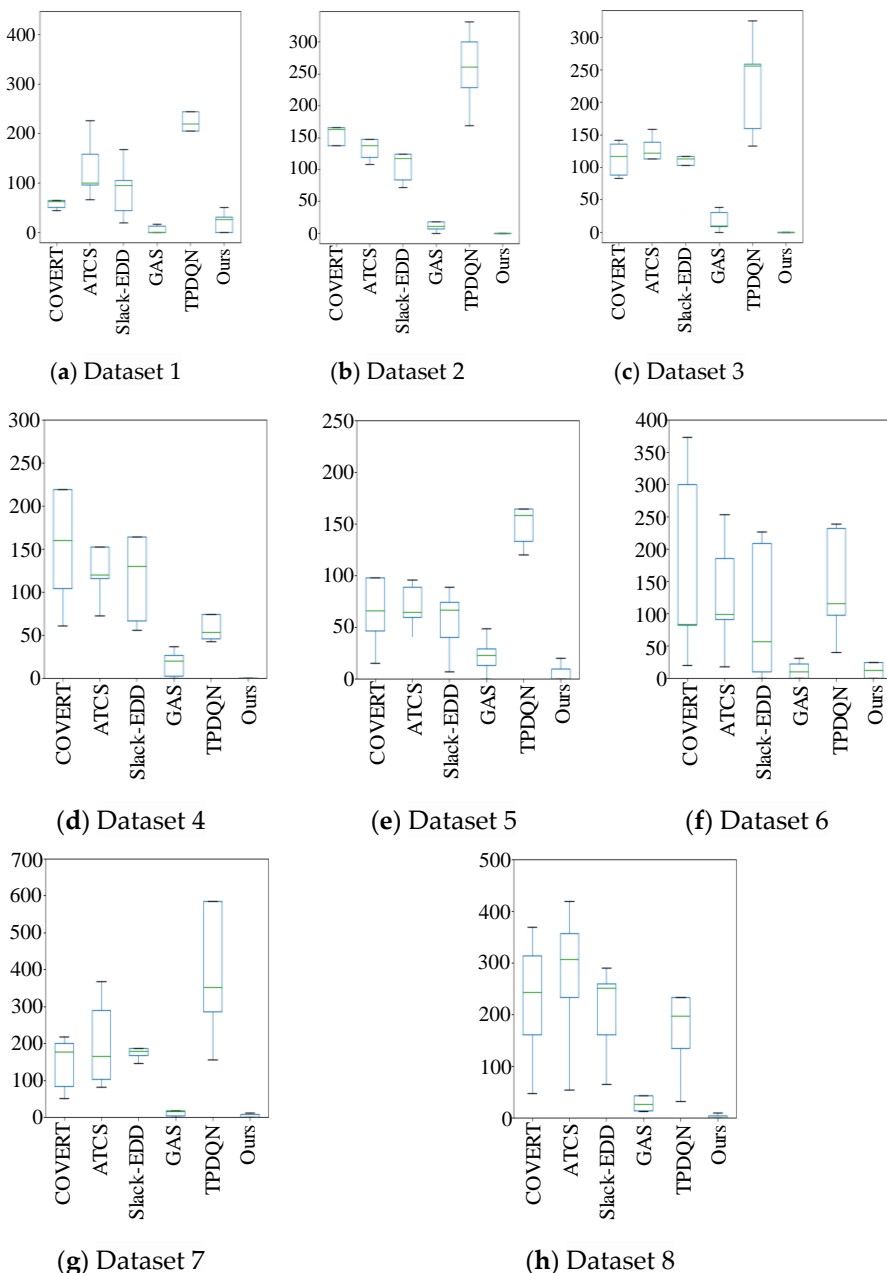

**Figure 4.** *RPD* obtained by all the schedulers for each dataset (horizontal axis: schedulers, vertical axis: *RPD*).

**Table 3.** Average *RPD* obtained by the schedulers for each dataset.

| Datasets | COVERT | ATCS | Slack-EDD | GAS | TPDQN |
|:---:|:---:|:---:|:---:|:---:|:---:|
| 1 | 75.71 | 128.98 | 85.46 | 5.70 | 237.01 |
| 2 | 152.36 | 141.17 | 125.09 | 13.88 | 257.84 |
| 3 | 113.15 | 117.94 | 101.88 | 17.36 | 226.22 |
| 4 | 191.89 | 190.05 | 162.45 | 16.78 | 80.02 |
| 5 | 85.14 | 68.11 | 55.36 | 22.74 | 161.21 |
| 6 | 171.81 | 129.39 | 100.44 | 12.46 | 144.81 |
| 7 | 146.00 | 201.15 | 208.91 | 36.17 | 555.95 |
| 8 | 226.93 | 273.95 | 205.54 | 38.96 | 253.291 |

Specifically, Figure 5 visualizes the improvement ratio of the proposed scheduler compared to GAS, TPDQN, and best rule presenting the best scheduling performances among the rules in terms of tardiness. For all the datasets, the DDQN scheduler enhances the scheduling performances by 155.605% compared to TPDQN in the best case of Dataset 7. In the worst case, the DDQN scheduler underperforms GAS by -12.21% in Dataset 1. In particular, Datasets 4 and 8 represent extra-large problems, the DDQN scheduler also better performs compared to TPDQN and GAS by 51.39% and 12.32%, 129.01% and 29.69%, respectively.

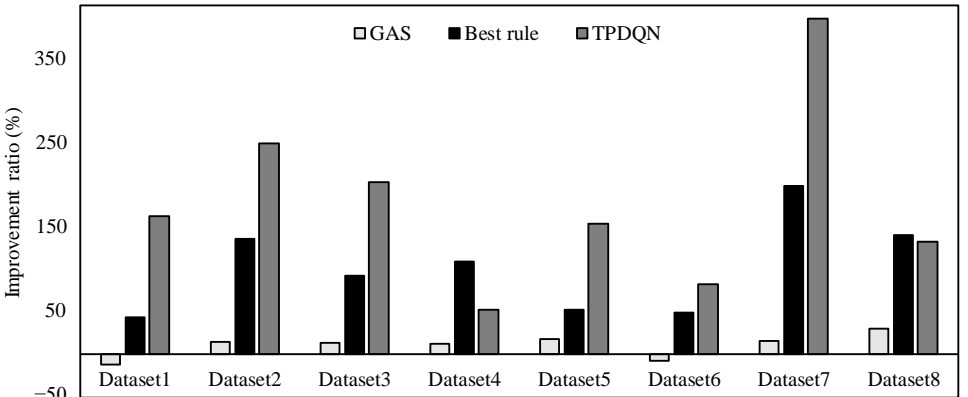

**Figure 5.** Improvement ratio of the proposed scheduler compared to GAS, TPDQN, and best rule in terms of tardiness.

It is interpreted that the proposed scheduler successfully finds the weight values of the Q-network able to determine one of the machines minimizing the tardiness of allocated jobs for PDMS problems through the proposed training phase [24]. The previous training strategy applied to TPDQN tends to fail to search for the effective weight values minimizing the tardiness of allocated jobs in the PDMS. GAS produces less tardiness for the small-size problem, but it is likely to be ineffective for large and extra-large problems since GAS requires more computational time when the size of problems is larger [42].

Finally, to compare the scheduling performances in terms of statistics, further experiments are conducted; we paired $t$-test with both the proposed scheduler and the conventional schedulers for each dataset. Table 4 depicts $\rho$-value results achieved by conducting the $t$-test between the proposed scheduler and the others considered at the 5% level of significance. The $\rho$-value of the proposed scheduler is less than 0.05 except for GAS against all datasets. Note that the DDQN scheduler significantly outperforms COVERT, ATCS, Slack-EDD, and TPDQN for all the datasets in terms of statistics, but it is better than GAS for some datasets in terms of statistics.

**Table 4.** $\rho$-values for tardiness difference between the proposed and the other schedulers.

| Datasets | COVERT | ATCS | Slack-EDD | GAS | TPDQN |
|---|---|---|---|---|---|
| 1 | $1.82^{-7}$ | $5.01^{-20}$ | $8.81^{-10}$ | 0.06 | $1.89^{-35}$ |
| 2 | $3.44^{-16}$ | $1.25^{-17}$ | $2.54^{-11}$ | 0.04 | $3.55^{-45}$ |
| 3 | $1.04^{-31}$ | $4.60^{-29}$ | $5.69^{-20}$ | 0.03 | $1.44^{-57}$ |
| 4 | $2.59^{-13}$ | $9.08^{-19}$ | $6.37^{-12}$ | 0.21 | $2.53^{-16}$ |
| 5 | $2.54^{-15}$ | $4.39^{-11}$ | $3.35^{-9}$ | $0.09^{-2}$ | $1.39^{-30}$ |
| 6 | $7.47^{-13}$ | $1.28^{-9}$ | $8.39^{-12}$ | 0.33 | $6.48^{-21}$ |
| 7 | $1.14^{-13}$ | $1.38^{-16}$ | $4.20^{-17}$ | 0.11 | $5.42^{-54}$ |
| 8 | $3.08^{-8}$ | $9.58^{-11}$ | $2.95^{-7}$ | 0.20 | $4.27^{-9}$ |

## 5. Discussion

To investigate the robustness of the proposed scheduler, eight datasets are prepared, as shown in Table 2. The conventional ones, such as Slack-EDD, ATCS, and COVERT, are

implemented. The meta-heuristic- and deep-Q-learning-based schedulers, named GAS and TPDQN, are additionally implemented, respectively. In terms of *RPD*, the proposed scheduler has the best performances for all the datasets, as presented in Figure 4 and Table 3. In particular, the proposed scheduler produces a schedule with quite less total tardiness for the large and extra-large problems, as shown in Figure 5. On the other hand, TPDQN shows ineffective scheduling performances for particular testing datasets. This might be related to the fact that it fails to find better Q-values for minimizing tardiness in the training phase. GAS gives effective schedules for small and medium scheduling problems, but it produces much tardiness for large and extra-large scheduling problems. It is interpreted that GAS is likely to fail in searching for better schedules within the limited computational time when the size of the scheduling problem grows. Finally, Table 4 depicts the results of the statistical analysis for each dataset. The results in Table 4 imply that the scheduling performances of the proposed scheduler are significant compared to the conventional schedulers except for GAS.

## 6. Conclusions

In this paper, we suggest a novel machine scheduler based on double deep Q-learning, called DDQN scheduler, to address a PDMS problem with sequence-independent setup time towards minimizing total tardiness. To avoid the allocation uncertainty of allocated jobs for dedicated machines and minimize the tardiness of allocated jobs, novel state, reward, and training algorithm are applied. The proposed scheduler successfully prevents the overestimation problem for the action values; this scheduler successfully acts an action for ensuring allocation certainty and minimizing the tardiness of allocated jobs.

An enhanced scheduling method able to provide a schedule for minimizing total tardiness within a short time has been required in modern manufacturing as effective scheduling results are directly related to the revenue of manufacturing companies [42]. Recently, many companies have been turning their attention to developing deep learning and reinforcement learning techniques. However, in the PDMS problem, the deep learning technique does not fit for the following two reasons. First, the deep learning technique requires huge training instances generated by optimized schedules, which are difficult to collect [28]. Further, it is very hard to design a relationship between a job and a machine as an output value.

From this viewpoint, reinforcement learning is more useful since it attempts to search for a better schedule through repeated trial and error processes in the training without training instances. In this study, reinforcement learning is adopted, and its scheduling performances are better than existing methods. Moreover, the proposed scheduler gives better schedules for the large and extra-large scheduling problems. This research provides a guideline for applying a reinforcement learning algorithm to the PDMS problem and encourages attempts to apply this algorithm in a practical environment.

The future direction is the following. When the dedicated machines change and the number of machines increases, re-training is necessary. To overcome this limitation, we plan to design a new state and action that is not necessary, re-training the Q-network. In addition, since the action is only designed to select a machine, the proposed scheduler may try to find a good schedule within a limited solution space. Thus, future work tries to redesign actions to consider all the possible pairs of jobs and machines. It also applies other methods, such as deep deterministic policy gradient and proximal policy optimization algorithms, to improve scheduling performances.

**Author Contributions:** Conceptualization, D.L. (Donghun Lee) and K.K.; methodology, D.L. (Donghun Lee), H.K., D.L. (Dongjin Lee) and J.L.; software and validation, H.K., D.L. (Dongjin Lee) and J.L.; formal analysis and investigation, D.L. (Donghun Lee), H.K. and K.K.; data curation, D.L. (Donghun Lee) and K.K.; writing—original draft preparation, D.L. (Donghun Lee) and K.K.; writing—review and editing, K.K.; project administration and funding acquisition, K.K.; All authors have read and agreed to the published version of the manuscript.

**Funding:** This work was supported by the Research Assistance Program (2020) in the Incheon National University and by the Korea Institute of Energy Technology Evaluation and Planning (KETEP) granted financial resource by the Ministry of Trade, Industry & Energy, Republic of Korea (No. 20212020900090).

**Conflicts of Interest:** The authors declare no conflict of interest.

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
