# Peer review of "Deep Reinforcement Learning-Based Scheduler on Parallel Dedicated Machine Scheduling Problem towards Minimizing Total Tardiness"

_sustainability, doi:10.3390/su15042920_

Round 1

Reviewer 1 Report

Comments to the authors:

1. Kindly avoid the term 'we' in the article.

2. In the introduction section, the research gap and novelty are not described.

3. Brief about the performance measures considered in the PMDS problem.

4. The authors selected only Total Tardiness. Why are other measures not considered?

5. What are the different deep learning and reinforcement models used in the published work?

6. The results are not discussed. A separate discussion section is required.

7 The proposed algorithm's performance is better. How?

8. The results needs to be discussed with literature support.

9. Rewrite the conclusion with research findings.

10. The developeed model can be used with any number of jobs and process.

11. Few literatures are too old. please update with recent papers.

Author Response

Please find the attached document regarding to the point-to-point responses for the reviewer.

Reviewer 2 Report

The authors have presented a nice work related to the deep Reinforcement Learning Based Scheduler on Parallel Dedicated Machine Scheduling Problem Towards Minimizing Total Tardiness. I have a few minor comments.

1. Please add some quantitative analysis in the abstract.

2. The novelty of the work should be more stressed in the last paragraph of the introduction section supported by some quantitative data.

3. Please provide reference for Equation (13).

Author Response

(The authors gave the same response as above.)

Reviewer 3 Report

The manuscript titled “Deep Reinforcement Learning Based Scheduler on Parallel Dedicated Machine Scheduling Problem Towards Minimizing Total Tardiness”, is the good attempt to adopt the reinforcement learning to solve the problem of job tardiness. The proposal shows the good results.

Specific Comments:

1. The literature review need to be improved by adding the articles from 2022.

2. Comparative analysis should have the comparison amongst the existing techniques.

3. In Figure 2, What are the dimensions of the various layers?

4. Once, check all the grammar errors in the manuscript.

It solves the problem of job tardiness wherein parallel dedicated machine scheduling is employed to minimize total tardiness for jobs consisting of a job type, processing time, and due date. In addition, this problem come under the category of NP-hard.

The proposed work is relevant and good contribution in the field. Moreover, it address the identified research gap.

The adoption of deep learning technique open the avenue for the improvement for the proposed work.

The proposed scheduler, repeatedly finds better Q-values towards minimizing tardiness of allocated jobs by updating the weights in a neural network.

The references are appropriate, only latest literature need to be added.

Author Response

(The authors gave the same response as above.)

Round 2

Reviewer 1 Report

All the best.